# COMPOSITIONAL KERNEL MACHINES

**Robert Gens & Pedro Domingos**
Department of Computer Science & Engineering
University of Washington
Seattle, WA 98195, USA
`{rcg,pedrod}@cs.washington.edu`

## ABSTRACT

Convolutional neural networks (convnets) have achieved impressive results on re-
cent computer vision benchmarks. While they benefit from multiple layers that en-
code nonlinear decision boundaries and a degree of translation invariance, training
convnets is a lengthy procedure fraught with local optima. Alternatively, a kernel
method that incorporates the compositionality and symmetry of convnets could
learn similar nonlinear concepts yet with easier training and architecture selec-
tion. We propose compositional kernel machines (CKMs), which effectively cre-
ate an exponential number of virtual training instances by composing transformed
sub-regions of the original ones. Despite this, CKM discriminant functions can
be computed efficiently using ideas from sum-product networks. The ability to
compose virtual instances in this way gives CKMs invariance to translations and
other symmetries, and combats the curse of dimensionality. Just as support vec-
tor machines (SVMs) provided a compelling alternative to multilayer perceptrons
when they were introduced, CKMs could become an attractive approach for object
recognition and other vision problems. In this paper we define CKMs, explore
their properties, and present promising results on NORB datasets. Experiments
show that CKMs can outperform SVMs and be competitive with convnets in a
number of dimensions, by learning symmetries and compositional concepts from
fewer samples without data augmentation.

## 1 INTRODUCTION

The depth of state-of-the-art convnets is a double-edged sword: it yields both nonlinearity for so-
phisticated discrimination and nonconvexity for frustrating optimization. The established training
procedure for ILSVRC classification cycles through the million-image training set more than fifty
times, requiring substantial stochasticity, data augmentation, and hand-tuned learning rates. On to-
day's consumer hardware, the process takes several days. However, performance depends heavily
on hyperparameters, which include the number and connections of neurons as well as optimization
details. Unfortunately, the space of hyperparameters is unbounded, and each configuration of hyper-
parameters requires the aforementioned training procedure. It is no surprise that large organizations
with enough computational power to conduct this search dominate this task.

Yet mastery of object recognition on a static dataset is not enough to propel robotics and internet-
scale applications with ever-growing instances and categories. Each time the training set is modified,
the convnet must be retrained ("fine-tuned") for optimum performance. If the training set grows
linearly with time, the total training computation grows quadratically.

We propose the Compositional Kernel Machine (CKM), a kernel-based visual classifier that has the
symmetry and compositionality of convnets but with the training benefits of instance-based learning
(IBL). CKMs branch from the original instance-based methods with *virtual instances*, an exponen-
tial set of plausible compositions of training instances. The first steps in this direction are promising
compared to IBL and deep methods, and future work will benefit from over fifty years of research
into nearest neighbor algorithms, kernel methods, and neural networks.

In this paper we first define CKMs, explore their formal and computational properties, and compare
them to existing kernel methods. We then propose a key contribution of this work: a sum-product
function (SPF) that efficiently sums over an exponential number of virtual instances. We then de-

scribe how to train the CKM with and without parameter optimization. Finally, we present results on NORB and variants that show a CKM trained on a CPU can be competitive with convnets trained for much longer on a GPU and can outperform them on tests of composition and symmetry, as well as markedly improving over previous IBL methods.

## 2 COMPOSITIONAL KERNEL MACHINES

The key issue in using an instance-based learner on large images is the curse of dimensionality. Even millions of training images are not enough to construct a meaningful neighborhood for a $256 \times 256$ pixel image. The compositional kernel machine (CKM) addresses this issue by constructing an exponential number of *virtual instances*. The core hypothesis is that a variation of the visual world can be understood as a rearrangement of low-dimensional pieces that have been seen before. For example, an image of a house could be recognized by matching many pieces from other images of houses from different viewpoints. The virtual instances represent this set of all possible transformations and recombinations of the training images. The arrangement of these pieces cannot be arbitrary, so CKMs learn how to compose virtual instances with weights on compositions. A major contribution of this work is the ability to efficiently sum over this set with a sum-product function.

The set of virtual instances is related to the nonlinear image manifolds described by Simard et al. (1992) but with key differences. Whereas the tangent distance accounts for transformations applied to the whole image, virtual instances can depict local transformations that are applied differently across an image. Secondly, the tangent plane approximation of the image manifold is only accurate near the training images. Virtual instances can easily represent distant transformations. Unlike the explicit augmentation of virtual support vectors in Schölkopf et al. (1996), the set of virtual instances in a CKM is implicit and exponentially larger. Platt & Allen (1996) demonstrated an early version of virtual instances to expand the set of negative examples for a linear classifier.

### 2.1 DEFINITION

We define CKMs using notation common to other IBL techniques. The two prototypical instance-based learners are $k$-nearest neighbors and support vector machines. The foundation for both algorithms is a similarity or kernel function $K(x, x')$ between two instances. Given a training set of $m$ labeled instances of the form $\langle x_i, y_i \rangle$ and query $x_q$, the $k$-NN algorithm outputs the most common label of the $k$ nearest instances:

$$y_{\text{kNN}}(x_q) = \arg \max_c \sum_{i=1}^{m} \mathbb{1}\left[c = y_i \wedge K(x_i, x_q) \geq K(x^k, x_q)\right]$$

where $\mathbb{1}[\cdot]$ equals one if its argument is true and zero otherwise, and $x^k$ is the $k^{\text{th}}$ nearest training instance to query $x_q$ assuming unique distances. The multiclass support vector machine (Crammer & Singer, 2001) in its dual form can be seen as a weighted nearest neighbor that outputs the class with the highest weighted sum of kernel values with the query:

$$y_{\text{SVM}}(x_q) = \arg \max_c \sum_{i=1}^{m} \alpha_{i,c} K(x_i, x_q) \tag{1}$$

where $\alpha_{i,c}$ is the weight on training instance $x_i$ that contributes to the score of class $c$.

The CKM performs the same classification as these instance-based methods but it sums over an exponentially larger set of virtual instances to mitigate the curse of dimensionality. Virtual instances are composed of rearranged elements from one or more training instances. Depending on the design of the CKM, elements can be subsets of instance variables (e.g., overlapping pixel patches) or features thereof (e.g., ORB features or a 2D grid of convnet feature vectors). We assume there is a deterministic procedure that processes each training or test instance $x_i$ into a fixed tuple of indexed elements $E_{x_i} = (e_{i,1}, \ldots, e_{i,|E_{x_i}|})$, where instances may have different numbers of elements. The query instance $x_q$ (with tuple of elements $E_{x_q}$) is the example that is being classified by the CKM; it is a training instance during training and a test instance during testing. A virtual instance $z$ is represented by a tuple of elements from training instances, e.g. $E_z = (e_{10,5}, e_{71,2}, \ldots, e_{46,17})$. Given a query instance $x_q$, the CKM represents a set of virtual instances each with the same number of elements as $E_{x_q}$. We define a leaf kernel $K_L(e_{i,j}, e_{i',j'})$ that measures the similarity between any two elements. Using kernel composition (Aronszajn, 1950), we define the kernel between the query instance $x_q$ and a virtual instance $z$ as the product of leaf kernels over their corresponding elements:

$K(z, x_q) = \prod_j^{|E_{x_q}|} K_L(e_{z,j}, e_{q,j}).$

We combine leaf kernels with weighted sums and products to compactly represent a sum over kernels with an exponential number of virtual instances. Just as a sum-product network can compactly represent a mixture model that is a weighted sum over an exponential number of mixture components, the same algebraic decomposition can compactly encode a weighted sum over an exponential number of kernels. For example, if the query instance is represented by two elements $E_{x_q} = (e_{q,1}, \ e_{q,2})$ and the training set contains elements $\{e_1, \ e_2, \ e_3, \ e_4, \ e_5, \ e_6\}$, then

$$[w_1 K_L(e_{q,1}, e_1) + w_2 K_L(e_{q,1}, e_2) + w_3 K_L(e_{q,1}, e_3)] \times$$
$$[w_4 K_L(e_{q,2}, e_4) + w_5 K_L(e_{q,2}, e_5) + w_6 K_L(e_{q,2}, e_6)]$$

expresses a weighted sum over nine virtual instances using eleven additions/multiplications instead of twenty-six for an expanded flat sum $w_1 K_L(e_{q,1}, e_1) K_L(e_{q,2}, e_4) + \ldots + w_9 K_L(e_{q,1}, e_3) K_L(e_{q,2}, e_6)$. If the query instance and training set contained 100 and 10000 elements, respectively, then a similar factorization would use $O(10^6)$ operations compared to a naïve sum over $10^{500}$ virtual instances. Leveraging the Sum-Product Theorem (Friesen & Domingos, 2016), we define CKMs to allow for more expressive architectures with this exponential computational savings.

**Definition 1.** *A compositional kernel machine (CKM) is defined recursively.*
  1. *A leaf kernel over a query element and a training set element is a CKM.*
  2. *A product of CKMs with disjoint scopes is a CKM.*
  3. *A weighted sum of CKMs with the same scope is a CKM.*

The scope of an operator is the set of query elements it takes as inputs; it is analogous to the receptive field of a unit in a neural network, but with CKMs the query elements are not restricted to being pixels on the image grid (e.g., they may be defined as a set of extracted image features). A leaf kernel has singleton scope, internal nodes have scope over some subset of the query elements, and the root node of the CKM has full scope of all query elements $E_{x_q}$. This definition allows for rich CKM architectures with many layers to represent elaborate compositions. The value of each sum node child is multiplied by a weight $w_{k,c}$ and optionally a constant cost function $\phi(e_{i,j}, e_{i',j'})$ that rewards certain compositions of elements. Analogous to a multiclass SVM, the CKM has a separate set of weights for each class $c$ in the dataset. The CKM classifies a query instance as $y_{\text{CKM}}(x_q) = \arg\max_c S_c(x_q)$, where $S_c(x_q)$ is the value of the root node of the CKM evaluating query instance $x_q$ using weights for class $c$.

**Definition 2** (Friesen & Domingos (2016)). *A product node is decomposable iff the scopes of its children are disjoint. An SPF is decomposable iff all of its product nodes are decomposable.*

**Theorem 1** (Sum-Product Theorem, Friesen & Domingos (2016)). *Every decomposable SPF can be summed over its domain in time linear in its size.*

**Corollary 1.** $S_c(x_q)$ *can sum over the set of virtual instances in time linear in the size of the SPF.*

*Proof.* For each query instance element $e_{q,j}$ we define a discrete variable $Z_j$ with a state for each training element $e_{i',j'}$ found in a leaf kernel $K_L(e_{q,j}, e_{i',j'})$ in the CKM. The Cartesian product of the domains of the variables $\mathbf{Z}$ defines the set of virtual instances represented by the CKM. $S_c(x_q)$ is a SPF over semiring $(R, \oplus, \otimes, 0, 1)$, variables $\mathbf{Z}$, constant functions $\mathbf{w}$ and $\boldsymbol{\phi}$, and univariate functions $K_L(e_{q,j}, Z_j)$. With the appropriate definition of leaf kernels, any semiring can be used. The definition above provides that the children of every product node have disjoint scopes. Constant functions have empty scope so there is no intersection with scopes of other children. With all product nodes decomposable, $S_c(x_q)$ is a decomposable SPF and can therefore sum over all states of $\mathbf{Z}$, the virtual instances, in time linear to the size of the CKM. $\qquad\square$

Special cases of CKMs include multiclass SVMs (flat sum-of-products) and naive Bayes nearest neighbor (Boiman et al., 2008) (flat product-of-sums). A CKM can be seen as a generalization of an image grammar (Fu, 1974) where terminal symbols corresponding to pieces of training images are scored with kernels and non-terminal symbols are sum nodes with a production for each child product node.

The weights and cost functions of the CKM control the weights on the virtual instances. Each virtual instance represented by the CKM defines a tree that connects the root to the leaf kernels over its unique composition of training set elements. If we were to expand the CKM into a flat sum (cf. Equation 1), the weight on a virtual instance would be the product of the weights and cost functions along the branches of its corresponding tree. These weights are important as they can prevent implausible virtual instances. For example, if we use image patches as the elements and allow all compositions, the set of virtual instances would largely contain nonsense noise patterns. If

the elements were pixels, the virtual instances could even contain arbitrary images from classes not present in the training set. There are many aspects of composition that can be encoded by the CKM. For example, we can penalize virtual instances that compose training set elements using different symmetry group transformations. We could also penalize compositions that juxtapose elements that disagree on the contents of their borders. Weights can be learned to establish clusters of elements and reward certain arrangements. In Section 3 we demonstrate one choice of weights and cost functions in a CKM architecture built from extracted image features.

## 2.2 LEARNING

The training procedure for a CKM builds an SPF that encodes the virtual instances. There are then two options for how to set weights in the model. As with $k$-NN, the weights in the CKM could be set to uniform. Alternatively, as with SVMs, the weights could be optimized to improve generalization and reduce model size.

For weight learning, we use block-coordinate gradient descent to optimize leave-one-out loss over the training set. The leave-one-out loss on a training instance $x_i$ is the loss on that instance made by the learner trained on all data except $x_i$. Though it is an almost unbiased estimate of generalization error (Luntz & Brailovsky, 1969), it is typically too expensive to compute or optimize with non-IBL methods (Chapelle et al., 2002). With CKMs, caching the SPFs and efficient data structures make it feasible to compute exact partial derivatives of the leave-one-out loss over the whole training set. We use a multiclass squared-hinge loss

$$\mathcal{L}(x_i, y_i) = \max \left[ 1 + \underbrace{S_{y'}(x_i)}_{\text{Best incorrect class}} - \underbrace{S_{y_i}(x_i)}_{\text{True class}}, \; 0 \right]^2$$

for the loss on training instance $x_i$ with true label $y_i$ and highest-scoring incorrect class $y'$. We use the squared version of the hinge loss as it performs better empirically and prioritizes updates to element weights that led to larger margin violations. In general, this objective is not convex as it involves the difference of the two discriminant functions which are strictly convex (due to the choice of semiring and the product of weights on each virtual instance). In the special case of the sum-product semiring and unique weights on virtual instances the objective is convex as is true for L2-SVMs. Convnets also have a non-convex objective, but they require lengthy optimization to perform well. As we show in Section 3, CKMs can achieve high accuracy with uniform weights, which further serves as good initialization for gradient descent.

For each epoch, we iterate through the training set, for each training instance $x_i$ optimizing the block of weights on those branches with $E_{x_i}$ as descendants. We take gradient steps to lower the leave-one-out loss over the rest of the training set $\sum_{i' \in ([1,m] \setminus i)} \mathcal{L}(x_{i'}, y_{i'})$. We iterate until convergence or an early stopping condition. A component of the gradient of the squared-hinge loss on an instance takes the form

$$\frac{\partial}{\partial w_{k,c}} \mathcal{L}(x_i, y_i) = \begin{cases} 2\Delta(x_i, y_i) \frac{\partial S_{y'}(x_i)}{\partial w_{k,c}} & \text{if } \Delta(x_i, y_i) > 0 \wedge c = y' \\ -2\Delta(x_i, y_i) \frac{\partial S_{y_i}(x_i)}{\partial w_{k,c}} & \text{if } \Delta(x_i, y_i) > 0 \wedge c = y_i \\ 0 & \text{otherwise} \end{cases}$$

where $\Delta(x_i, y_i) = 1 + S_{y'}(x_i) - S_{y_i}(x_i)$. We compute partial derivatives $\frac{\partial S_c(x_i)}{\partial w_{k,c}}$ with backpropagation through the SPF. For efficiency, terms of the gradient can be set to zero and excluded from backpropagation if the values of corresponding leaf kernels are small enough. This is either exact (e.g., if $\oplus$ is maximization) or an approximation (e.g., if $\oplus$ is normal addition).

## 2.3 SCALABILITY

CKMs have several scalability advantages over convnets. As mentioned previously, they do not require a lengthy training procedure. This makes it much easier to add new instances and categories. Whereas most of the computation to evaluate a single setting of convnet hyperparameters is sunk in training, CKMs can efficiently race hyperparameters on hold-out data (Lee & Moore, 1994).

The evaluation of the CKM depends on the structure of the SPF, the size of the training set, and the computer architecture. A basic building block of these SPFs is a sum node with a number of children on the order of magnitude of the training set elements $|\mathcal{E}|$. On a sufficiently parallel

Table 1: Dataset properties

| Name | #Training Exs. - #Testing Exs. | Dimensions | Classes |
|---|---|---|---|
| Small NORB | 24300-24300 | $96 \times 96$ | 5 |
| NORB Compositions | 100-1000 | $256 \times 256$ | 2 |
| NORB Symmetries | $\{50, 100, \ldots, 12800\}$-2916 | $108 \times 108$ | 6 |

computer, assuming the size of the training set elements greatly exceeds the dimensionality of the leaf kernel, this sum node will require $O(\log(|\mathcal{E}|))$ time (the depth of a parallel $\oplus$ reduction circuit) and $O(|\mathcal{E}|)$ space. Duda et al. (2000) describe a constant time nearest neighbor circuit that relies on precomputed Voronoi partitions, but this has impractical space requirements in high dimensions. As with SVMs, optimization of sparse element weights can greatly reduce model size.

On a modest multicore computer, we must resort to using specialized data structures. Hash codes can be used to index raw features or to measure Hamming distance as a proxy to more expensive distance functions. While they are perhaps the fastest method to accelerate a nearest neighbor search, the most accurate hashing methods involve a training period yet do not necessarily result in high recall (Torralba et al., 2008; Heo et al., 2012). There are many space-partitioning data structure trees in the literature, however in practice none are able to offer exact search of nearest neighbors in high dimensions in logarithmic time. In our experiments we use hierarchical $k$-means trees (Muja & Lowe, 2009), which are a good compromise between speed and accuracy.

## 3 EXPERIMENTS

We test CKMs on three image classification scenarios that feature images from either the small NORB dataset or the NORB jittered-cluttered dataset (LeCun et al., 2004). Both NORB datasets contain greyscale images of five categories of plastic toys photographed with varied altitudes, azimuths, and lighting conditions. Table 1 summarizes the datasets. We first describe the SPN architecture and then detail each of the three scenarios.

### 3.1 EXPERIMENTAL ARCHITECTURE

In our experiments the architecture of the SPF $S_c(x_q)$ for each query image is based on its unique set of extracted ORB features. Like SIFT features, ORB features are rotation-invariant and produce a descriptor from intensity differences, but ORB is much faster to compute and thus suitable for real time applications (Rublee et al., 2011). The elements $E_{x_i} = (e_{i,1}, \ldots, e_{i,|E_i|})$ of each image $x_i$ are its extracted keypoints, where an element's feature vector and image position are denoted by $\vec{f}(e_{i,j})$ and $\vec{p}(e_{i,j})$ respectively. We use the max-sum semiring ($\oplus = \max$, $\otimes = +$) because it is more robust to noisy virtual instances, yields sparser gradients, is more efficient to compute, and performs better empirically compared with the sum-product semiring.

The SPF $S_c(x_q)$ maximizes over variables $\boldsymbol{Z} = (Z_1, \ldots, Z_{|E_{x_q}|})$ corresponding to query elements $E_{x_q}$ with states for all possible virtual instances. The SPF contains a unary scope max node for every variable $\{Z_j\}$ that maximizes over the weighted kernels of all possible training elements $\mathcal{E}$: $\oplus(Z_j) = \bigoplus_{z_j \in \mathcal{E}} w_{z_j,c} \otimes K_L(z_j, e_{q,j})$. The SPF contains a binary scope max node for all pairs of variables $\{Z_j, Z_{j'}\}$ for which at least one corresponding query element is within the $k$-nearest spatial neighbors of the other. These nodes maximize over the weighted kernels of all possible combinations of training set elements.

$$\oplus(Z_j, Z_{j'}) = \bigoplus_{z_j \in \mathcal{E}} \bigoplus_{z_{j'} \in \mathcal{E}} w_{z_j,c} \otimes w_{z_{j'},c} \otimes \phi(z_j, z_{j'}) \otimes K_L(z_j, e_{q,j}) \otimes K_L(z_{j'}, e_{q,j'}) \quad (2)$$

This maximizes over all possible pairs of training set elements, weighting the two leaf kernels by two corresponding element weights and a cost function. We use a leaf kernel for image elements that incorporates both the Hamming distance between their features and the Euclidean distance between their image positions: $K_L(e_{i,j}, e_{i',j'}) = \max(\beta_0 - \beta_1 d_{\text{Ham}}(\vec{f}(e_{i,j}), \vec{f}(e_{i',j'})), 0) + \max(\beta_2 ||(\vec{p}(e_{i,j}), \vec{p}(e_{i',j'})||, \beta_3)$. This rewards training set elements that look like a query instance element and appear in a similar location, with thresholds for efficiency. This can represent, for example, the photographic bias to center foreground objects or a discriminative cue from seeing sky at the top of the image. We use the pairwise cost function $\phi(e_{i,j}, e_{i',j'}) = \mathbb{1}[i = i']\beta_4$ that rewards combinations of elements from the same source training image. This captures the intuition that

compositions sourced from more images are less coherent and more likely to contain nonsense than those using fewer. The image is represented as a sum of these unary and binary max nodes. The scopes of children of the sum are restricted to be disjoint, so the children $\{\oplus(Z_1, Z_2), \oplus(Z_2, Z_3)\}$ would be disallowed, for example. This restriction is what allows the SPF to be tractable, and with multiple sums the SPF has high-treewidth. By comparison, a Markov random field expressing these dependencies would be intractable. The root max node of the SPF has $P$ sums as children, each of which has its random set of unary and binary scope max node children that cover full scope $\mathbf{Z}$. We illustrate a simplified version of the SPF architecture in Figure 1. Though this SPF models limited image structure, the definition of CKMs allows for more expressive architectures as with SPNs.

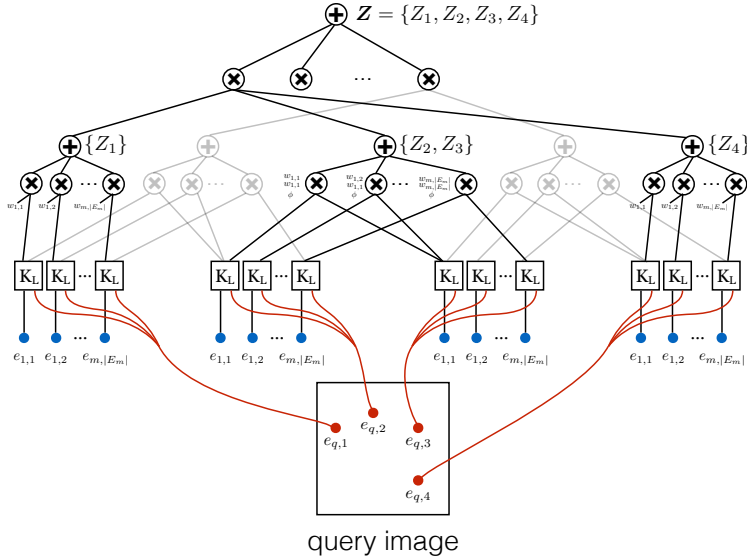

query image

Figure 1: Simplified illustration of the SPF $S_c(x_q)$ architecture with max-sum semiring used in experiments (using ORB features as elements, $|E_{x_q}| \approx 100$). Red dots depict elements $E_{x_q}$ of query instance $x_q$. Blue dots show training set elements $e_{i,j} \in \mathcal{E}$, duplicated with each query element for clarity. A boxed $K_L$ shows the leaf kernel with lines descending to its two element arguments. The $\oplus$ nodes are labeled with their scopes. Weights and cost functions (arguments omitted) appear next to $\otimes$ nodes. Only a subset of the unary and binary scope $\oplus$ nodes are drawn. Only two of the $P$ top-level $\otimes$ nodes are fully detailed (the children of the second are drawn faded).

In the following sections, we refer to two variants **CKM** and **CKM$_W$**. The **CKM** version uses uniform weights $w_{k,c}$, similar to the basic $k$-nearest neighbor algorithm. The **CKM$_W$** method optimizes weights $w_{k,c}$ as described in Section 2.2. Both versions restrict weights for class $c$ to be $-\infty$ ($\oplus$ identity) for those training elements not in class $c$. This constraint ensures that method **CKM** is discriminative (as is true with $k$-NN) and reduces the number of parameters optimized by **CKM$_W$**. The hyperparameters of ORB feature extraction, leaf kernels, cost function, and optimization were chosen using grid search on a validation set.

With our CPU implementation, **CKM** trains in a single pass of feature extraction and storage at $\sim$5ms/image, **CKM$_W$** trains in under ten epochs at $\sim$90ms/image, and both versions test at $\sim$80ms/image. The GPU-optimized convnets train at $\sim$2ms/image for many epochs and test at $\sim$1ms/image. Remarkably, **CKM** on a CPU trains faster than the convnet on a GPU.

### 3.2 SMALL NORB

We use the original train-test separation which measures generalization to new instances of a category (i.e. tested on toy truck that is different from the toys it was trained on). We show promising results in Table 2 comparing CKMs to deep and IBL methods. With improvement over $k$-NN and SVM, the **CKM** and **CKM$_W$** results show the benefit of using virtual instances to combat the curse of dimensionality. We note that the **CKM** variant that does not optimize weights performs nearly as well as the **CKM$_W$** version that does. Since the test set uses a different set of toys, the use of untrained ORB features hurts the performance of the CKM. Convnets have an advantage here because they discriminatively train their lowest level of features and represent richer image structure in their architecture. To become competitive, future work should improve upon this preliminary CKM

Table 2: Accuracy on Small NORB

| Method | Accuracy |
|---|---|
| Convnet (14 epochs) (Bengio & LeCun, 2007) | 94.0% |
| DBM with aug. training (Salakhutdinov & Hinton, 2009) | 92.8% |
| **CKM**$_W$ | 89.8% |
| Convnet (2 epochs) (Bengio & LeCun, 2007) | 89.6% |
| DBM (Salakhutdinov & Hinton, 2009) | 89.2% |
| SVM (Gaussian kernel) (Bengio & LeCun, 2007) | 88.4% |
| **CKM** | 88.3% |
| $k$-NN (LeCun et al., 2004) | 81.6% |
| Logistic regression (LeCun et al., 2004) | 77.5% |

Table 3: Accuracy on NORB Compositions

| Method | Accuracy | Train+Test (min) |
|---|---|---|
| **CKM** | 82.4% | 1.5 [CPU] |
| SVM with convnet features | 75.0% | 1 [GPU+CPU] |
| Convnet | 50.6% | 9 [GPU] |
| $k$-NN on image pixels | 51.2% | 0.2 [CPU] |

architecture. We demonstrate the advantage of CKMs for representing composition and symmetry in the following experiments.

## 3.3 NORB COMPOSITIONS

A general goal of representation learning is to disentangle the factors of variation of a signal without having to see those factors in all combinations. To evaluate progress towards this, we created images containing three toys each, sourced from the small NORB training set. Small NORB contains ten types of each toy category (e.g., ten different airplanes), which we divided into two collections. Each image is generated by choosing one of the collections uniformly and for each of three categories (person, airplane, animal) randomly sampling a toy from that collection with higher probability ($P = \frac{5}{6}$) than from the other collection (i.e., there are two children with disjoint toy collections but they sometimes borrow). The task is to determine which of the two collections generated the image. This dataset measures whether a method can distinguish different compositions without having seen all possible permutations of those objects through symmetries and noisy intra-class variation. Analogous tasks include identifying people by their clothing, recognizing social groups by their members, and classifying cuisines by their ingredients.

We compare CKMs to other methods in Table 3. Convnets and their features are computed using the TensorFlow library (Abadi et al., 2015). Training convnets from few images is very difficult without resorting to other datasets; we augment the training set with random crops, which still yields test accuracy near chance. In such situations it is common to train an SVM with features extracted by a convnet trained on a different, larger dataset. We use 2048-dimensional features extracted from the penultimate layer of the pre-trained Inception network (Szegedy et al., 2015) and a linear kernel SVM with squared-hinge loss (Pedregosa et al., 2011). Notably, the CKM is much more accurate than the deep methods, and it is about as fast as the SVM despite not taking advantage of the GPU.

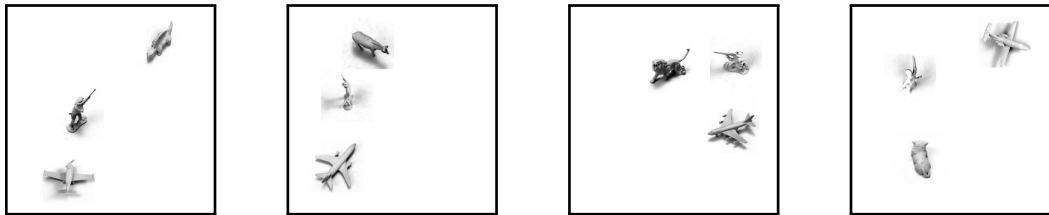

Figure 2: Images from NORB Compositions

## 3.4 NORB SYMMETRIES

Composition is a useful tool for modeling the symmetries of objects. When we see an image of an object in a new pose, parts of the image may look similar to parts of images of the object in poses we have seen before. In this experiment, we partition the training set of NORB jittered-cluttered into a

new dataset with 10% withheld for each of validation and testing. Training and testing on the same group of toy instances, this measures the ability to generalize to new angles, lighting conditions, backgrounds, and distortions.

We vary the amount of training data to plot learning curves in Figure 3. We observe that CKMs are better able to generalize to these distortions than other methods, especially with less data. Importantly, the performance of **CKM** improves with more data, without requiring costly optimization as data is added. We note that the benefit of $\mathbf{CKM}_W$ using weight learning becomes apparent with 200 training instances. This learning curve suggests that CKMs would be well suited for applications in cluttered environments with many 3D transformations (e.g., loop closure).

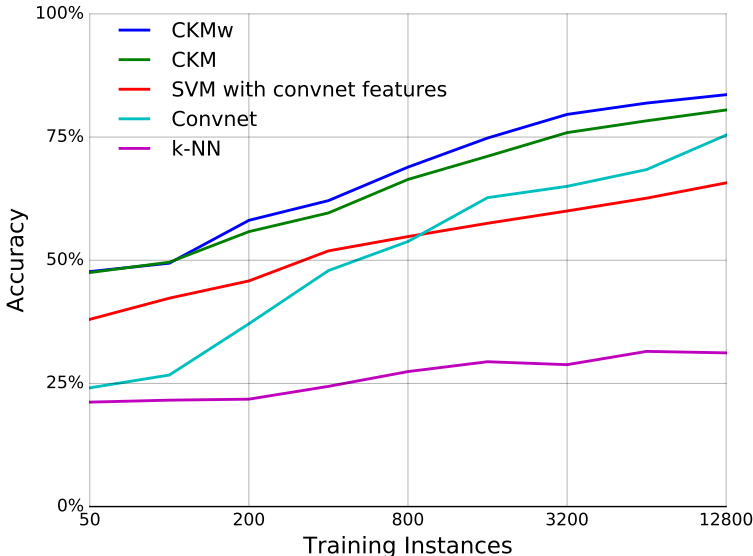

Figure 3: Number of training instances versus accuracy on unseen symmetries in NORB

## 4 CONCLUSION

This paper proposed compositional kernel machines, an instance-based method for object recognition that addresses some of the weaknesses of deep architectures and other kernel methods. We showed how using a sum-product function to represent a discriminant function leads to tractable summation over the weighted kernels to an exponential set of virtual instances, which can mitigate the curse of dimensionality and improve sample complexity. We proposed a method to discriminatively learn weights on individual instance elements and showed that this improves upon uniform weighting. Finally, we presented results in several scenarios showing that CKMs are a significant improvement for IBL and show promise compared with deep methods.

Future research directions include developing other architectures and learning procedures for CKMs, integrating symmetry transformations into the architecture through kernels and cost functions, and applying CKMs to structured prediction, regression, and reinforcement learning problems. CKMs exhibit a reversed trade-off of fast learning speed and large model size compared to neural networks. Given that animals can benefit from both trade-offs, these results may inspire computational theories of different brain structures, especially the neocortex versus the cerebellum (Ito, 2012).

ACKNOWLEDGMENTS

The authors are grateful to John Platt for helpful discussions and feedback. This research was partly supported by ONR grant N00014-16-1-2697, AFRL contract FA8750-13-2-0019, a Google PhD Fellowship, an AWS in Education Grant, and an NVIDIA academic hardware grant. The views and conclusions contained in this document are those of the authors and should not be interpreted as necessarily representing the official policies, either expressed or implied, of ONR, AFRL, or the United States Government.

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
