# Peer review of "Compositional Kernel Machines"

_ICLR 2017 — rejected_

[Official Review · AnonReviewer3 · rating 5 · confidence 4 · 15 Dec 2016]
**interesting idea, but too preliminary**

This paper proposes a new learning framework called "compositional kernel machines" (CKM). It combines two ideas: kernel methods and sum-product network (SPN). CKM first defines leaf kernels on elements of the query and training examples, then it defines kernel recursively (similar to sum-product network). This paper has shown that the evaluation CKM can be done efficiently using the same tricks in SPN.

Positive: I think the idea in this paper is interesting. Instance-based learning methods (such as SVM with kernels) have been successful in the past, but have been replaced by deep learning methods (e.g. convnet) in the past few years. This paper investigate an unexplored area of how to combine the ideas from kernel methods and deep networks (SPN in this case). 

Negative: Although the idea of this paper is interesting, this paper is clearly very preliminary. In its current form, I simply do not see any advantage of the proposed framework over convnet. I will elaborate below.

1) One of the most important claims of this paper is that CKM is faster to learn than convnet. I am not clear why that is the case. Both CKM and convnet use gradient descent during learning, why would CKM be faster?

Also during inference, the running time of convnet only depends on its network structure. But for CKM, in addition to the network structure, it also depends on the size of training set. From this perspective, it does not seem CKM is very scalable when the training size is big. That is probably why this paper has to use all kinds of specialized data structures and tricks (even on a fairly simple dataset like NORB)

2) I am having a hard time understanding what the leaf kernel is capturing. For example, if the "elements" correspond to raw pixel intensities, a leaf kernel essentially compares the intensity value of a pixel in the query image with that in a training image. But in this case, wouldn't you end up comparing a lot of background pixels across these two images (which does not help with recognition)?

I think it probably helps to explain Sec 3.1 a bit better. In its current form, this part is very dense and hard to understand.

3) It is also not entirely clear to me how you would design the architecture of the sum-product function. The example is Sec 3.1 seems to be fairly arbitrary.

4) The experiment section is probably the weakest part. NORB is a very small and toy-ish dataset by today's standard. Even on this small dataset, the proposed method is only slighly better than SVM (it is not clear whether "SVM" in Table 2 is linear SVM or kernel SVM. If it is linear SVM, I suspect the performance of "SVM" will be even higher when you use kernel SVM), and far worse than convnet. The proposed method only shows improvement over convnet on synthetic datasets (NORB compositions, NORM symmetries)

Overall, I think this paper has some interesting ideas. But in its current form, it is a bit too preliminary and more work is needed to show its advantage. Having said that, I acknowledge that in the machine learning history, many important ideas seem pre-mature when they were first proposed, and it took time for these ideas to develop.

[Official Review · AnonReviewer1 · rating 6 · confidence 4 · 22 Dec 2016]
**An alternative to convolutional neural networks (early stage)**

Thank you for an interesting read. The ideas presented have a good basis of being true, but the experiments are rather too simple. It would be interesting to see more empirical evidence.

Pros
- The approach seems to decrease the training time, which is of prime importance in deep learning. Although, that comes at a price of slightly more complex model.
- There is a grounded theory for sum-product functions which is basis for the compositional architecture described in the paper. Theoretically, any semiring and kernel could be used for the model which decreases need for handcrafting the structure of the model, which is a big problem in existing convolutional neural networks.

Cons
- The experiments are on very simple dataset NORB. Although, it is great to understand a model's dynamics on a simpler dataset, some analysis on complex datasets are important to act as empirical evidence. The compositional kernel approach is compared to convolutional neural networks, hence it is only fair to compare said results on large datasets such as Imagenet.

Minor
- Section 3.4 claims that CKMs model symmetries of objects. It felt that ample justification was not provided for this claim

[Official Review · AnonReviewer5 · rating 5 · confidence 4 · 31 Dec 2016]
**Interesting but much more detail is needed**

The authors propose a method to efficiently augment an SVM variant with many virtual instances, and show promising preliminary results. The paper was an interesting read, with thoughtful methodology, but has partially unsupported and potentially misleading claims.

Pros:
- Thoughtful methodology with sensible design choices
- Potentially useful for smaller (n < 10000) datasets with a lot of statistical structure
- Nice connections with sum-product literature

Cons:
- Claims about scalability are very unclear
- Generally the paper does not succeed in telling a complete story about the properties and applicability of the proposed method.
- Experiments are very preliminary 

The scalability claims are particularly unclear. The paper repeatedly mentions lack of scalability as a drawback for convnets, but it appears the proposed CKM is less scalable than a standard SVM, yet SVMs often handle much fewer training instances than deep neural networks. It appears the scalability advantages are mostly for training sets with roughly fewer than 10,000 instances -- and even if the method could scale to >> 10,000 training instances, it's unclear whether the predictive accuracy would be competitive with convnets in that domain. Moreover, the idea of doing 10^6 operations simply for creating virtual instances on 10^4 training points and 100 test points is still somewhat daunting. What if we had 10^6 training instances and 10^5 testing instances?  Because scalability (in the number of training instances) is one of the biggest drawbacks of using SVMs (e.g. with Gaussian kernels) on modern datasets, the scalability claims in this paper need to be significantly expanded and clarified. On a related note, the suggestion that convnets grow quadratically in computation with additional training instances in the introduction needs to be augmented with more detail, and is potentially misleading. Convnets typically scale linearly with additional training data. 

In general, the paper suffers greatly from a lack of clarity and issues of presentation. As above, the full story is not presented, with critical details often missing. Moreover, it would strengthen the paper to remove broad claims such as "Just as support vector machines (SVMs) eclipsed multilayer perceptrons in the 1990s, CKMs could become a compelling alternative to convnets with reduced training time and sample complexity", suggesting that CKMs could eclipse convolutional neural networks, and instead provide more helpful and precise information. Convnets are multilayer perceptrons used in the 1990s (as well as now) and they are not eclipsed by SVMs -- they have different relative advantages. And based on the information presented, broadly advertising scalability over convnets is misleading. Can CKMs scale to datasets with millions of training and test instances?  It seems as if the scalability advantages are limited to smaller datasets, and asymptotic scalability could be much worse in general. And even if CKMs could scale to such datasets would they have as good predictive accuracy as convnets on those applications? Being specific and with full disclosure about the precise strengths and limitations of the work would greatly improve this paper.

CKMs may be more robust to adversarial examples than standard convnets, due to the virtual instances. But there are many approaches to make deep nets more robust to adversarial examples. It would be useful to consider and compare to these. The ideas behind CKMs also are not inherently specific to kernel methods. Have you considered looking at using virtual instances in a similar way with deep networks? A full exploration might be its own paper, but the idea is worth at least brief discussion in the text. 

A big advantage of SVMs (with Gaussian kernels) over deep neural nets is that one can achieve quite good performance with very little human intervention (design choices). However, CKMs seem to require extensive intervention, in terms of architecture (as with a neural network), and in insuring that the virtual instances are created in a plausible manner for the particular application at hand. It's very unclear in general how one would want to create sensible virtual instances and this topic deserves further consideration. Moreover, unlike SVMs (with for example Gaussian or linear kernels) or standard convolutional networks, which are quite general models, CKMs as applied in this paper seem more like SVMs (or kernel methods) which have been highly tailored to a particular application -- in this case, the NORB dataset. There is certainly nothing wrong with the tailored approach, but it would help to be clear and detailed about where the presented ideas can be applied out of the box, or how one would go about making the relevant design choices for a range of different problems. And indeed, it would be good to avoid the potentially misleading suggestions early in the paper that the proposed method is a general alternative to convnets.

The experiments give some insights into the advantages of the proposed approach, but are very limited. To get a sense of the properties --the strengths and limitations -- of the proposed method, one needs a greater range of datasets with a much larger range of training and test sizes. The comparisons are also quite limited: why not an SVM with a Gaussian kernel?  What about an SVM using convnet features from the dataset at hand (light blue curve in figure 3) -- it should do at least as well as the light blue curve. There are also other works that could be considered which combine some of the advantages of kernel methods with deep networks. Also the claim that the approach helps with the curse of dimensionality is sensible but not particularly explored. It also seems the curse of dimensionality could affect the scalability of creating a useful set of virtual instances. And it's unclear how CKM would work without any ORB features. 

Even if the method can (be adapted to) scale to n >> 10000, it's unclear whether it will be more useful than convnets in that domain. Indeed, in the experiments here, convnets essentially match CKMs in performance after 12,000 examples, and would probably perform better than CKMs on larger datasets.  We can only speculate because the experiments don't consider larger problems.

The methodology largely takes inspiration from sum product networks, but its application in the context of a kernel approach is reasonably original, and worthy of exploration. It's reasonable to expect the approach to be significant, but its significance is not demonstrated.

The quality is high in the sense that the methods and insights are thoughtful, but suffers from broad claims and a lack of full and precise detail.

In short: I like the paper, but it needs more specific details, and a full disclosure of where the method should be most applicable, and its precise advantages and limitations.  Code would be helpful for reproducibility.

[Official Review · AnonReviewer2 · rating 5 · confidence 3 · 02 Jan 2017]
**No Title**

This paper proposes a new learning model "Compositional Kernel Machines (CKMs)" that extends the classic kernel machines by constructing compositional kernel functions using sum-product networks. This paper considers the convnets as nicely learned nonlinear decision functions and resort their success in classification to their compositional nature. This perspective motivates the design of compositional kernel functions and the sum-product implementation is indeed interesting. I agree the composition is important for convnets, but it is not the whole story of convnets' success. One essential difference between convnets and CKMs is that all the kernels in convnets are learned directly from data while CKMs still build on top of feature descriptors. This, I believe, limits the representation power of CKMs. A recent paper "Deep Convolutional Networks are Hierarchical Kernel Machines" by Anselmi, F. et al. seems to be interesting to the authors.
Experiments seem to be preliminary in this paper. It's good to see promising results of CKMs on small NORB, but it is quite important to show competitive results on recent classification standard benchmarks, such as MNIST, CIFAR10/100 and even Imagenet, in order to establish a novel learning model. In NORB compositions, CKMs seem to be better than convnets at classifying images by their dominant objects. I suspect it is because the use of sparse ORB features. It will be great if this paper could show the accuracy of ORB features with matching kernel SVMs. Some details about this experiment need further clarification, such as what are the high and low probabilities of sampling from each collections and how many images are generated. In NORB Symmetries, CKMs show better performance than convnets with small data, but the convnets seem not converged yet. Could it be possible to show results with larger dataset?

[Final Decision · Program Chairs · 06 Feb 2017]
**ICLR committee final decision**

There is consensus among the reviewers that the proposed method has potential merit, but that the experimental evaluation is too preliminary to warrant publication of the current manuscript. The paper also appears to make broad claims that are not fully supported by the results of the study. I encourage the authors to address the comments of the reviewers in future revisions of this work. Meanwhile, this paper would make a good contribution to the workshop track.